# Layer-Parallel Training for Transformers

## Abstract

We present a new training methodology for transformers using a multilevel layer-parallel approach. Through a neural ODE formulation of transformers, our application of a multilevel parallel-in-time algorithm for the forward and backpropagation phases of training achieves parallel acceleration over the layer dimension. This dramatically enhances parallel scalability as the network depth increases, particularly useful in large foundational models. However, achieving this introduces errors that cause systematic bias in the gradients, which in turn reduces convergence when closer to the minima. We develop algorithms to detect this critical transition and either switch to serial training, or systematically increase the accuracy of layer-parallel training. Results, including BERT, GPT, ViT, and machine translation architectures, demonstrate parallel-acceleration as well as accuracy commensurate with serial pre-training while fine-tuning is unaffected.

## 1 Introduction

The transformer (Vaswani et al., 2017) is an attention-based, sequence-to-sequence model, variations of which have achieved cutting-edge results in many areas, including natural language processing, computer vision, audio processing, and multi-modality; it serves as the backbone for foundation models (Bodnar et al., 2024; Zhou et al., 2024). Its performance surpasses RNNs due to parallelization across the sequence dimension, as well as access to all prior positions inside its context window. The network depth is limited by computational cost and memory footprint, however, increasing depth is one strategy for enhancing accuracy in language processing (Wang et al., 2024).

The challenge when parallelizing over depth is that transformers are based on a sequence of residual layers whose forward and backward propagations are inherently serial. This serialization is also endemic to neural ODE models that treat depth (layers) as the time-dimension. However, the scientific computing community has proposed a solution to serial ODE simulations by developing parallel-in-time methods (Gander, 2015; Ong and Schroder, 2020), including multigrid-reduction-in-time (MGRIT) (Falgout et al., 2014), which is used here. These techniques decompose the time domain of the discretized ODE, and develop iterative strategies to *simultaneously* solve over all layers.

To this end, we develop an MGRIT-based strategy for transformer training that exposes parallelism over the layer dimension, resulting in greater potential speedup as the depth increases. We use a neural ODE transformer formulation, which enables the application of parallel-in-time forward and backward propagation schemes. Our novel contributions are:

- Layer-parallel training (Gunther et al., 2020) is applied to transformers with increasing depth yielding speedups and reduced per-device memory overhead on multiple GPUs.
- Layer-parallel training creates inexact gradients with bias. Methodologies for detecting when the bias becomes too large, and remedying through reversion to a serial algorithm are shown. The overall process preserves significant parallel speedup.
- Detailed studies are shown for different parameter choices in the layer-parallel training algorithm in terms of pre-training performance and overall training accuracy, as well as fine-tuning on language model data sets.
- To the best of our knowledge, this is the first work to develop a neural ODE formulation for an encoder–decoder transformer architecture and to demonstrate its application.

## 2 Background

**Transformers.** Transformers have revolutionized natural language processing (NLP) since their introduction in 2017 (Vaswani et al., 2017). They leverage self-attention, which computes the relevance of each token in the input sequence to every other token. This facilitates the capture of long-range dependencies and contextual information, thereby enhancing the model's ability to understand and generate human language. This architecture mitigates the vanishing gradient problem, and reduces training time by enabling efficient utilization of modern hardware accelerators. However, transformers have trended towards increasingly deeper architectures. While early models like BERT (Devlin et al., 2018) and GPT (Radford et al., 2018) featured a modest number of layers, typically around 6 to 24, recent advancements have seen a dramatic increase in depth. For instance, GPT-3 employs 96 layers, while deeper architectures have been explored in T5 and Switch Transformer models (Xue et al., 2020; Fedus et al., 2022; Wang et al., 2024). Increasing the number of layers has been instrumental in achieving state-of-the-art performance across a range of NLP tasks, including machine translation, text summarization, and question answering. Thus, the transformer scalability is critical with large and deep models taking longer to train.

**Parallelization Techniques.** Parallelization in machine learning has been well-studied (Ben-Nun and Hoefler, 2019; Narayanan et al., 2021; Danieli et al., 2023). The most common approach is *data parallelism* (Valiant, 1990), with robust implementations in existing libraries (Dean et al., 2012; Abadi et al., 2016; Paszke et al., 2019). Data parallelism replicates the model across multiple GPUs and distributes the training data, allowing each to process a subset of the data simultaneously. This parallelization mitigates the cost of large data sets and batch-sizes. However, larger models demand additional parallelism due to memory requirements and gradient storage. This is addressed by *model parallelism* that splits the model across multiple GPUs. This enables training models whose memory requirements exceeds that of a single GPU. Implementations, often referred to as tensor parallelism, that mitigate computational costs with increased network width can be found in (Bradbury et al., 2018; Rasley et al., 2020). Combining model and data parallelism yields further speedups.

A final form is *pipeline parallelism* tries to mitigate growth in network depth by distributing layers across processors (Huang et al., 2019; Rasley et al., 2020). Here, batches are further subdivided into minibatches and then streamed through the layers in a pipeline fashion. By carefully rearranging work in an asynchronous way, parallel speedups can be obtained and the memory overhead is naturally distributed across multiple GPUs. For instance Li et al. (2021b) applies pipeline parallelism along the sequence length of decoder-only transformers and proposes an algorithm that dynamically finds the optimal sequence partitions to equally share the computational load among devices. In Zhuang et al. (2023), the authors study the optimal combination of inter-op parallelism (pipeline parallelism) and intra-op parallelism (tensor parallelism). While Korthikanti et al. (2023) uses tensor and sequence parallelism to store activations, thus avoiding their recomputation.

The layer-parallel approach has the same goal as pipelining, to mitigate the computational cost of growing network depth, while avoiding some of the its drawbacks. Pipelining works by over-decomposing the data and relying on the distribution of layers to achieve parallelism. This risks giving up some data parallelism (Narayanan et al., 2019), which typically has excellent scalability. Moreover, pipeline acceleration is reduced if the number of layers is much greater than the length of the pipeline due to computational latency, or the bubble phenomenon. The layer-parallel algorithm, on the other hand, *introduces controllable errors in forward and backward propagation in exchange for additional parallelism*. This yields an iterative algorithm that is fully compatible with data and model parallelism.

**Multigrid in time.** Multigrid methods are iterative solvers for large, sparse (non)linear systems (Trottenberg et al., 2001; Brandt and Livne, 2011; Hackbusch, 2013). The most well-known are geometric multigrid methods applied to elliptic partial differential equations (PDEs) in space. These methods utilize a hierarchy of grids and restriction/prolongation operators to efficiently reduce high-frequency errors on finer grids with relaxation (e.g., Jacobi or Gauss-Seidel) and low-frequency errors on coarser grids. The effectiveness and near optimal algorithmic complexity has been demonstrated across various physical domains.

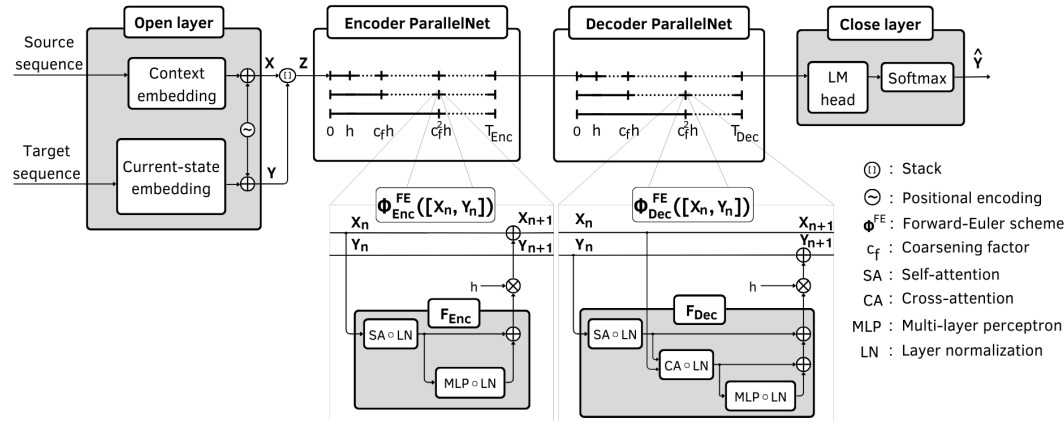

Figure 1: Layer-parallel transformer. The ParallelNet contains a time grid hierarchy with the coarsening rate denoted by $c_f$. Experiments use a fine level time-step of $h = 1$.

Multigrid is extended to the time dimension by the MGRIT algorithm (Falgout et al., 2014; Dobrev et al., 2017). By recasting serial time stepping as a single, global space-time operator, temporal multigrid methods employ a similar hierarchical approach to address errors. This approach accelerates time-dependent simulations, particularly for problems involving long time horizons or in systems where other parallelization dimensions are saturated.

The layer-parallel algorithm, proposed in Gunther et al. (2020), uses MGRIT to exploit the time dependent nature of neural ODE models. Layer-parallel applies an iterative MGRIT method to the forward and backward propagation phases to compute an *inexact* gradient. Through a partition of the layer dimension, this form of model-parallelism distributes the model across many devices, allowing arbitrarily large models, provided enough GPUs are available. Layer-parallel training can be used in conjunction with data and other forms of model parallelism. While layer-parallel has been used in other architectures (Cyr et al., 2019; Sun et al., 2021; Moon and Cyr, 2022), this is the first time it has been applied to transformers.

## 3 Layer-Parallel Transformers

### 3.1 ODE formulation of Transformers

In this work, we consider transformer architectures (e.g., BERT (Devlin et al., 2018), MarianNMT (Junczys-Dowmunt et al., 2018)) with pre-layer normalization (Xiong et al., 2020). Forward propagation through the encoder, with $N_{\text{Enc}}$ layers, is given as

$$\mathbf{X}_{n+1} = \mathbf{X}_n + h \underbrace{\left(\varphi_1\left(\mathbf{X}_n, \boldsymbol{\theta}_{n,1}\right) + \varphi_2\left(\mathbf{X}_n + \varphi_1\left(\mathbf{X}_n, \boldsymbol{\theta}_{n,1}\right), \boldsymbol{\theta}_{n,2}\right)\right)}_{:=\boldsymbol{F}_{\text{Enc}}(t_n, \mathbf{X}_n)}, \tag{1}$$

where $\mathbf{X}_n$ is the $n$-th layer input, $\mathbf{X}_{n+1}$ is the $n$-th layer output, and $h$ is typically $h = 1$. The symbol $\boldsymbol{\theta}_{n,j}$ denotes the parameters of the $n$-th layer's $j$-th sublayer, which is parametrized by evaluating $\boldsymbol{F}_{\text{Enc}}$ at $t_n$. The functions $\varphi_1$ and $\varphi_2$ are defined as $\varphi_1 := \text{SA} \circ \text{LN}$, and $\varphi_2 := \text{MLP} \circ \text{LN}$, where SA, LN, and MLP denote self-attention, layer norm, and MLP respectively. Decoder-only architectures (e.g., GPT (Radford et al., 2018)) are similar, with the addition of a causal mask in the attention.

In case of encoder-decoder architectures, the decoder, with $N_{\text{Dec}}$ layers, has the form:

$$\mathbf{Y}_{n+1} = \mathbf{Y}_n + h \underbrace{\left(\varphi_1(\mathbf{Y}_n, \boldsymbol{\theta}_{n,1}) + \varphi_3(\mathbf{Y}_n + \varphi_1(\mathbf{Y}_n, \boldsymbol{\theta}_{n,1}), \mathbf{X}_{N_{\text{Enc}}}, \boldsymbol{\theta}_{n,3}) + \varphi_2(\overline{\mathbf{Y}}_n, \boldsymbol{\theta}_{n,2})\right)}_{:=\boldsymbol{F}_{\text{Dec}}(t_n, \mathbf{Y}_n, \mathbf{X}_{N_{\text{Enc}}})}, \tag{2}$$

**Two-level MGRIT:**
1. Parallel FCF-relaxation with $\mathbf{S}_0$
2. Restrict $\mathbf{r}_0 = \mathbf{G}_0 - \mathbf{A}_0\mathbf{W}_0$
   to coarse-level, yielding $\mathbf{r}_1$
3. Solve for coarse-level error $\mathbf{e}_1$
   serially with $\mathbf{A}_1\mathbf{e}_1 = \mathbf{r}_1$
4. Correct fine-level solution
   with $\mathbf{e}_1$

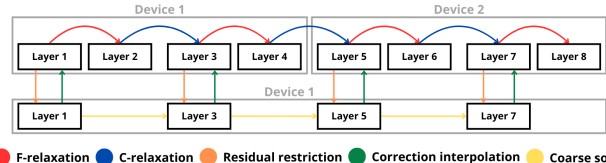

Figure 2: 2-level MGRIT pseudocode (left), MGRIT with $c_f = 2$, $L = 2$ on 2 devices (right).

where $\overline{\mathbf{Y}}_n = \mathbf{Y}_n + \varphi_1(\mathbf{Y}_n, \boldsymbol{\theta}_{n,1}) + \varphi_3(\mathbf{Y}_n + \varphi_1(\mathbf{Y}_n, \boldsymbol{\theta}_{n,1}), \mathbf{X}_{N_{\text{Enc}}}, \boldsymbol{\theta}_{n,3})$. The symbol $\boldsymbol{\theta}_n$ denotes the parameters of $n$-th decoder layer, $\mathbf{Y}_n$ denotes the $n$-th decoder-layer input, and $\mathbf{X}_{N_{\text{Enc}}}$ denotes the encoder output. The function $\varphi_3$ is $\varphi_3 := \text{CA} \circ \text{LN}$, where CA is cross-attention.

To facilitate application of parallel-in-time, we associate the transformer architecture with a neural ODE. The idea of viewing the forward propagation as ODE discretizations was first introduced for ResNets by E (2017), establishing a continuous-depth perspective on deep networks. The stability and well-posedness of the forward propagation was then studied in Haber and Ruthotto (2017); Lu et al. (2018); Chen et al. (2018), where the authors apply several numerical schemes to train deep neural networks. This ODE-based formulation was later expanded to encoder-only transformers in Queiruga et al. (2021) and Li et al. (2021a). In this work, we further extend the formulation to encoder-decoder transformer architectures.

To this aim, consider a mapping $\boldsymbol{\theta}$ defined on $[0, T]$ which interpolates $\boldsymbol{\theta}_n$ at $t_n$, where $T := T_{\text{Enc}} + T_{\text{Dec}}$, $T_{\text{Enc}} := hN_{\text{Enc}}$, and $T_{\text{Dec}} := hN_{\text{Dec}}$. For encoder-decoder transformers, we stack the encoder states $\mathbf{X}$ and the (shifted) decoder states $\mathbf{Y}$, such that the forward propagation through the transformer is defined as

$$\mathbf{Z}_{n+1} := [\mathbf{X}_{n+1}, \mathbf{Y}_{n+1-N_{\text{Enc}}}] = \mathbf{Z}_n + h\,\mathbf{F}(t_n, \mathbf{Z}_n), \tag{3}$$

where $\mathbf{X}_n := \mathbf{X}_{N_{\text{Enc}}}$, $\forall n > N_{\text{Enc}}$, and $\mathbf{Y}_n := \mathbf{Y}_0$, $\forall n < N_{\text{Enc}}$. The function $\mathbf{F}$ is given as

$$\mathbf{F}(t, [\mathbf{X}, \mathbf{Y}], ) := \begin{cases} [\boldsymbol{F}_{\text{Enc}}(t, \mathbf{X}), & \mathbf{0} & ], & \text{if } t < T_{\text{Enc}}, \\ [\mathbf{0} & , & \boldsymbol{F}_{\text{Dec}}(t, \mathbf{Y}, \mathbf{X})], & \text{if } t \geq T_{\text{Enc}}. \end{cases}$$

For encoder-only transformers, $T := T_{\text{Enc}}$, $\mathbf{Z} := \mathbf{X}$, and $\boldsymbol{F} := \boldsymbol{F}_{\text{Enc}}$.

Thus, we can interpret the transformer forward propagation (3) as a forward Euler discretization with time step $h = 1$ of the initial value problem (IVP) in eq. (4) (left). Here, for a given time $t \in [0, T]$, $\boldsymbol{F}$ depends on the states $\mathbf{Z}(t)$ and parameters $\boldsymbol{\theta}(t)$. The initial value $\mathbf{Z}_0$ is defined analogously on $\mathbf{X}_0$ and $\mathbf{Y}_0$, which denote the positionally-encoded source and target embeddings, respectively. Next, the gradients required for training can be then obtained by solving the adjoint equation in eq. (4) (right) backward in time. Here, $\mathscr{L}$ is the loss and $\boldsymbol{\lambda}(t_n)$ is the backpropagated gradients at the $n$-th layer.

$$\text{forward:} \begin{cases} \frac{d\mathbf{Z}}{dt} = \boldsymbol{F}(\mathbf{Z}, \boldsymbol{\theta}) \\ \mathbf{Z}(0) = \mathbf{Z}_0 \end{cases} \qquad \text{backward:} \begin{cases} \frac{\partial \boldsymbol{\lambda}}{\partial t} = \boldsymbol{\lambda}^T \frac{\partial \mathbf{F}}{\partial \mathbf{Z}} \\ \boldsymbol{\lambda}(t_N) = \frac{\partial \mathscr{L}}{\partial \mathbf{Z}(t_N)} \end{cases} \tag{4}$$

### 3.2 MGRIT: INEXACT FORWARD AND BACKWARD PROPAGATION

MGRIT constructs a hierarchy of discretizations of the IVP (4) (left). Starting with a fine time-step size $h$ on level 0, progressively coarser discretizations are constructed with a coarsening factor $c_f$ (e.g., the first coarse-level has step size $c_f h$, the second coarse-level $c_f^2 h$, and so on). The coarser levels correct the fine-grid solution, accelerating convergence to the serial solution. The fine-grid is evaluated only locally, in a highly parallel manner.

### 3.2.1 INEXACT MGRIT FORWARD PROPAGATION

For the MGRIT hierarchy, let $l = 0 \ldots L-1$ for $L$ total levels, then define $\mathbf{A}_l \mathbf{W}_l = \mathbf{G}_l$ where

$$\mathbf{A}_l = \begin{bmatrix} I & & & \\ (-I - c_f^l h \mathbf{F}_1) & I & & \\ & \ddots & \ddots & \\ & & (-I - c_f^l h \mathbf{F}_{N_l-1}) & I \end{bmatrix}, \quad \mathbf{W}_l = \begin{bmatrix} \mathbf{Z}_1 \\ \mathbf{Z}_2 \\ \vdots \\ \mathbf{Z}_{N_l} \end{bmatrix}, \quad \mathbf{G}_l = \begin{bmatrix} (I + c_f^l h \mathbf{F}_0) \mathbf{Z}_0 \\ \mathbf{0} \\ \vdots \\ \mathbf{0} \end{bmatrix},$$

and number of time-steps $N_l = N/c_f^l$. For $l = 0$ this is the iterative evolution Eq. 3 written as a system. The variable $c_f$ is a user-defined integer coarsening factor (often 2 or 4). The initial condition is implicitly included in the first entry of $\mathbf{G}_l$. The application of the nonlinear operator $\mathbf{A}_l$ written in matrix notation is understood as component-wise nonlinear composition (e.g. $\mathbf{F}_k \mathbf{Z}_k = \mathbf{F}_k(\mathbf{Z}_k)$).

*Remark:* The exact solution to the system when $l = 0$ yields the same state vector $\mathbf{W}_0$ as would be obtained from forward propagation of the neural ODE transformer. Further, the solution to the system $l + 1$ is an $O(h)$ approximation of the system at $l$.

Figure 2 outlines the algorithmic approach used by a 2-level MGRIT method ($L = 2$) to exploit this hierarchy to solve $\mathbf{A}_0 \mathbf{W}_0 = \mathbf{G}_0$. The first step of this algorithm, see Fig. 2 (left), applies a smoothing operator $\mathbf{S}_l \approx \mathbf{A}_l^{-1}$ on the fine grid. This smoother is selected to reduce high frequency errors in a parallel way (Dobrev et al., 2017). A form of block Jacobi, FCF-relaxation (fine-coarse-fine), is the approach taken in layer-parallel and is described in detail in Appendix A and Gunther et al. (2020). For level $l$, the application of FCF has $N_l/c_f^l$ way parallelism. Figure 2 (right) indicates this parallel relaxation phase with the red and blue arrows. The red arrows execute concurrently, followed by concurrent execution of the the blue arrows. After relaxation the error has been reduced locally over subsets of layers, yet no end to end communication has occurred.

Step 2 in Fig. 2 computes the residual $\mathbf{r}_0 = \mathbf{G}_0 - \mathbf{A}_0 \mathbf{W}_0$ on the fine grid and then "restricts" the residual with injection to the coarse level (orange arrows), yielding $\mathbf{r}_1$. In the third step, the "coarse solve" step (yellow arrows) computes the error on the coarse-level implied by the residual by solving $\mathbf{A}_1 \mathbf{e}_1 = \mathbf{r}_1$ exactly. This communicates end-to-end across the domain in serial, albeit at a factor of $c_f$ cheaper than the fine grid (the number of time steps is $N_0/c_f$ on level 1). In Step 4, this error correction is "interpolated" back up to the fine grid (green arrows). The algorithm repeats as required until a stopping criteria is met.

One iteration of this algorithm is referred to as a V-cycle. To create a hierarchy with more levels, the serial coarse solve can be replaced with another two-level solve and the whole process proceeds recursively. The serial coarsest-level solve will then be $c_f^{L-1}$ times cheaper than the fine grid, with additional parallel relaxation work done on intermediate levels. Critical for layer-parallel performance is that only a handful of V-cycle iterations are needed for sufficient accuracy, which results in approximate forward or backward propagation.

To initialize MGRIT for the system $\mathbf{A}_0$, we distribute the layers across multiple GPUs. In Figure 2 (right), an example is shown, where an 8 layer network is split over 2 GPUs/processors; a coarsening factor of $c_f = 2$ is shown. The example shows the second GPU stores $\mathbf{F}_4$ through $\mathbf{F}_7$, and an initial guess for $\mathbf{X}_4$. GPU-aware MPI is used for inter-device communication. See Cyr et al. (2025) for more implementation details. Similar to model parallelism, layer-parallel distributes the network across multiple GPUs, reducing the per device memory requirement.

### 3.2.2 INEXACT MGRIT BACKWARD PROPAGATION

To evaluate the gradients in parallel, the same MGRIT algorithm can be applied to solve the discretized adjoint problem (4) (right) backward in time; see (Gunther et al., 2020; Cyr et al., 2025) for details. Notably, in many cases, a single MGRIT iteration for the adjoint problem is enough to approximate the gradient with sufficient accuracy, enabling significant speedups. This behavior is consistent with findings in the literature, which indicate that optimizer convergence is significantly more sensitive to noise in the loss function evaluations than in gradient evaluations (Bellavia et al., 2023; Lou et al., 2025). As a result, the MGRIT

forward solve typically requires more iterations than backward. In the results section, forward iterations will refer to the number of MGRIT iterations used for forward propgation, and backward iterations for backward propagation, which will typically be smaller.

### 3.2.3 Adaptive control of the inexactness

Statistically biased error from inexact gradient evaluations is known to change the convergence properties of stochastic gradient descent algorithms. However, theory indicates that this can be mitigated if the error can be controlled as the minima is approached (Lin, 2022; Demidovich et al., 2023). Thus, detecting when the error is too large relative to the gradient is crucial for the application of corrective measures such as increasing the number of iterations or switching to exact solves. Due to the dynamics of transformers, MGRIT may require too many iterations to obtain a sufficient speedup relative to serial. To address this, we monitor the effectiveness of MGRIT iterations during training by evaluating the "convergence factor", defined as the ratio of consecutive fine-level residuals for iteration $k$, $\|\mathbf{r}_0^{(k+1)}\|/\|\mathbf{r}_0^{(k)}\|$. A small convergence factor implies rapid convergence. To ensure robustness, we periodically, every few (e.g. 500) batches, double the number of MGRIT iterations to monitor the convergence factor of the final iteration. A convergence factor above 1 indicates that the iteration count is no longer effective. The mitigation either improves the accuracy by increasing iteration count, or switching to serial training. Our results confirm, as suggested by the biased SGD theory (Demidovich et al., 2023), that despite the initial phase using inexact gradients improving accuracy in later stages leads to a network with comparable performance.

## 4    Numerical Results

To evaluate the efficacy of layer-parallel training and inference, we consider the following networks and applications, with additional details provided in the appendix.

1. **BERT pre-training** is the classical language modeling training problem for a pure encoder only network. The training objectives are the next sentence prediction (NSP) and masked-language modeling (MLM), however we only utilize MLM learning (Liu et al., 2019). For the pre-training data, we utilize the C4 dataset (Raffel et al., 2020).

2. **Morphological classification (MC)** is associated with classifying a word to its morphological class (noun, adjective, adverb, etc). We use the GUM corpus (Zeldes, 2017) dataset from Universal Dependencies (Nivre et al., 2017) and employ the neural ODE encoder-only transformer architecture, specified in Queiruga et al. (2021).

3. **Vision transformer (ViT)** is an encoder-only image transformer that is applied to tokens constructed from sub-partitions of an image (Dosovitskiy et al., 2020). We apply a classical ViT modified to be a neural ODE to the ImageNet dataset (Deng et al., 2009).

4. **Machine translation (MT)** consists of translating German sentences into English using the OPUS data set (Tiedemann and Thottingal, 2020), the pre-trained MarianTokenizer (Tiedemann and Thottingal, 2020), and an encoder-decoder transformer inspired by Junczys-Dowmunt et al. (2018) with the neural ODE modifications from above.

5. **GPT2 pre-training** is the decoder-only language model developed by OpenAI (Radford et al., 2018). We use the nanoGPT implementation (Karpathy, 2022) trained on OpenWebText (Gokaslan and Cohen, 2019) with minor modifications to the time stepping detailed in appendix B.

For each task, we demonstrate that layer-parallel forward and backward propagation, with adaptive control of inexactness, achieves the same accuracy as serial computations. We further show the strong scalability properties with respect to a varying number of transformer blocks $N$, the MGRIT coarsening factor $c_f$, and the number of MGRIT levels $L$. Hyperparameter configurations for all benchmark problems are provided in the appendix.

### 4.1    Convergence of MGRIT

In this section, we demonstrate the impact of the layer-parallel approach on training accuracy.

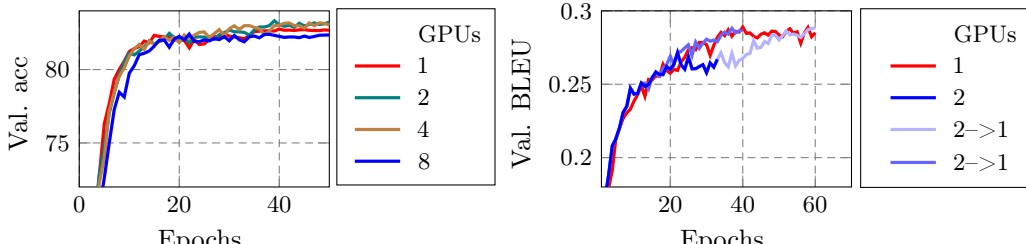

Figure 3: The long term training behavior using sequential and using layer-parallel with multiple GPUs. On the left, the validation accuracy for the MC example with 64 transformer layers, $L = 2$, and $c_f = 2$. On the right, the validation BLEU for the MT example with 6-6 transformer layers, $L = 2$, and $c_f = 3$. The plot corresponding to "2–>1" label illustrates a switch from parallel training with 2 GPUs to serial training with 1 GPU. Note that two depicted "2–>1" runs switch off from a parallel run at different points during the training.

**MC Training**   Figure 3 (left) compares the behavior of sequential and layer-parallel training with increasing number of GPUs. The inexactness in the gradients does not negatively impact the validation accuracy: layer-parallel achieves the same accuracy as serial training.

**MT Training**   Figure 3 (right) illustrates how the error accumulated due to inexact loss and gradient evaluations can lead to a slight deterioration in validation BLEU compared to the serial baseline. However, switching to sequential training after an efficient, parallel phase, allows the optimizer to quickly recover the validation BLEU score achieved in serial.

**BERT/GPT/ViT Pretraining**   In Figure 4 (left), we show the loss value of pretraining a 128 layer BERT model (Wang et al., 2024) using serial (blue), pure layer-parallel (red) and switching from parallel to serial (green, with multiple seeds shaded in grey). The layer-parallel configuration is 2 levels for both forward and backward solve, and $c_f = 4$ on 4 GPUs. We use this large model as an exemplar for the loss dynamics. During pretraining, layer-parallel converges past the loss plateau typical of BERT (Nagatsuka et al., 2021; Fu et al., 2023), but diverges and then stagnates. The divergence is expected as near local minima, smaller gradients can be overwhelmed by inexact gradient error.

We can use the indicator as described in Section 3.2.3 to demarcate the need to switch to using serial (exact) gradient computations. Figure 4 (left, green) shows after switching training dynamics closely match the serial case. In Table 1, performance differences are displayed for a few GLUE benchmarks comparing serial trained models, to those trained with the switching training schemes. The differences between the fine-tuned models demonstrate layer-parallel yields parallel speedups and commensurate accuracy.

The loss results for GPT and ViT follows an extremely similarly trajectory, as shown in Figure 4 (middle/right). While GPT is a decoder-only network, and ViT operates on image data, we see that inexact gradients (red) causes a divergence in training dynamics compared to exact dynamics (blue). However, by using the indicator, we are able to recover the dynamics by switching from serial to parallel where the indicator Figure 5 dictates. We use a 32 layer ViT with the neural ODE modification with the serial forward, and one level parallel backwards for MGRIT using 2 GPUs. The GPT network consists of 20 layers with the neural ODE modification on only the middle 16 layers with serial forward and one level parallel backwards; for more detail, please see Appendix B.

Table 1: Absolute differences in loss and accuracy of subset of GLUE task performance comparison between serial and parallel followed by serial (adaptive switching)

| Task Name | Diff. in Loss | Diff. in Accuracy |
| --- | --- | --- |
| CoLA (Corpus of Linguistic Acceptability) | 3.99e-4 | 0% |
| MRPC (Microsoft Research Paraphrase Corpus) | 1.10e-2 | 0% |
| QNLI (Question Natural Language Inference) | 3.38e-4 | 1.2% |

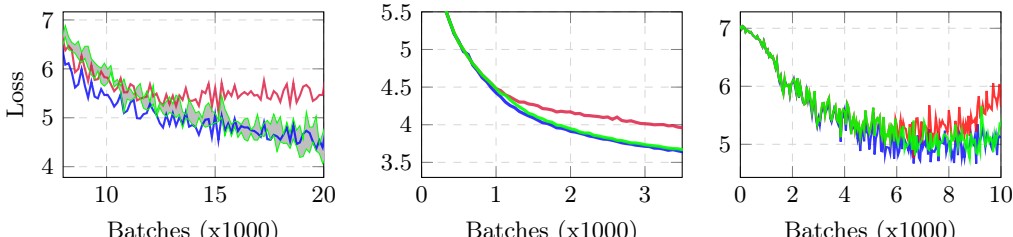

Figure 4: Plots of the loss for serial (blue), pure parallel (red) and switching to serial from parallel (green) for the BERT (left), GPT (middle) and ViT (right). In all the experiments, we see that purely layer parallel runs will diverge from serial training after a certain point. However, one can recover the original dynamics by switching from parallel to serial at an appropriate time given by the indicator. The gray color in the BERT subplot indicates the min/max over three different seeds.

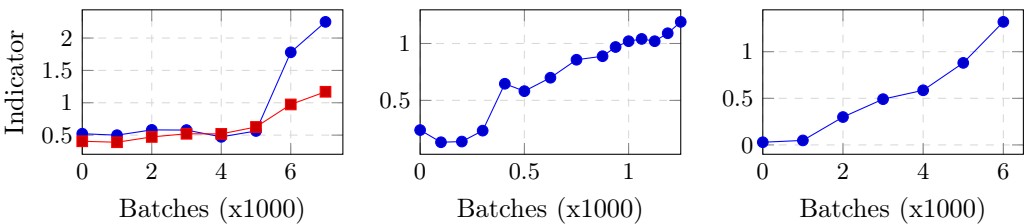

Figure 5: The indicator values for BERT (forward in red, backward in blue), ViT and GPT (backward in blue) using MGRIT. We see that at the 70000th batch, 1000th, and 6000th batch respectively, the indicators exceed 1, meaning that one should switch to exact gradient computation then.

## 4.2 SCALING STUDIES

In this section, we investigate the parallel scaling properties of the layer-parallel approach. Figure 6 shows the speedup achieved for encodere-only transforms on the: (left) the BERT task with $c_f = 4$, (middle) the MC task with $c_f = 2$, and (right) the ViT model with $c_f = 4$. All tasks use $L = 2$ levels. The obtained results indicate that the numerical and communication overhead introduced by MGRIT may occasionally lead to increased execution time when using two GPUs for small problems. However, as more computational resources are are employed for deeper models, the layer-parallelism enabled by MGRIT yields a substantial reduction in the overall execution time.

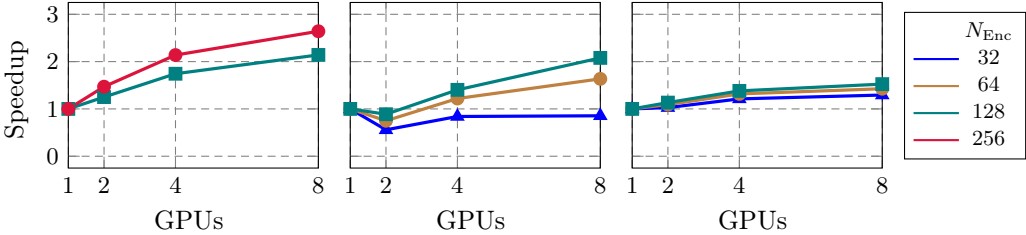

Figure 6: Speedup of layer-parallel for encoder-only transformers using $L = 2$. Left: BERT, on Singra, 1 forward, and 1 backward iteration, with $c_f = 4$. Middle: MC, on Jean-Zay, 2 forward, 1 backward iterations, with $c_f = 2$. Right: ViT, on Singra, serial forward, and 1 backward iteration, with $c_f = 4$. See appendix C for system details.

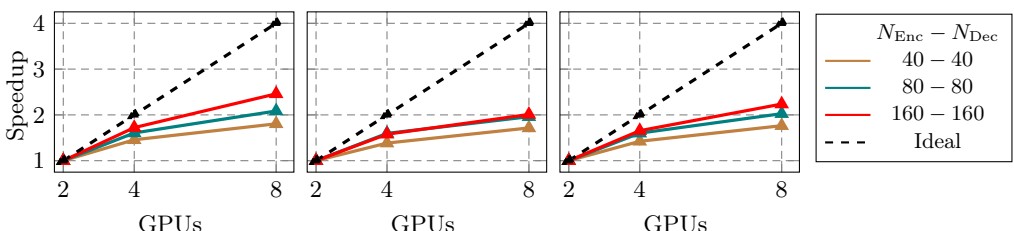

Figure 7: Strong scaling on Jean-Zay with respect to increasing number of layers $N_{\text{Enc}} + N_{\text{Dec}}$ for MT task. MGRIT uses $c_f = 4$, $L = 2$, 1 backward, and 2 forward iterations.

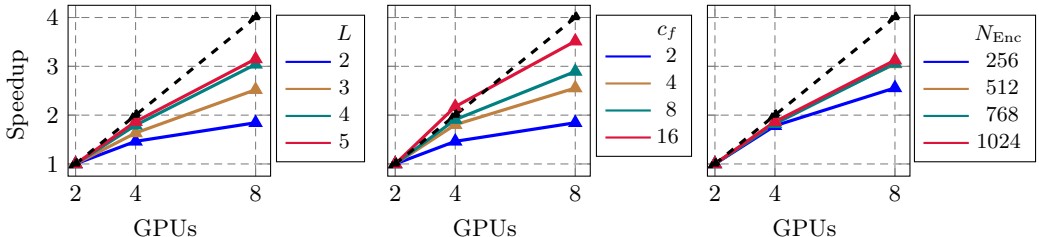

Figure 8: The impact of MGRIT parameters on scaling properties. The experiment is performed using 2 forward and 1 backward iteration for the MC task on Jean-Zay. Left: $c_f = 2$ and $N_{\text{Enc}} = 1024$. Middle: $L = 2$ and $N_{\text{Enc}} = 1024$. Right: $L = 3$ and $c_f = 4$. The blank line depicts ideal scaling.

A similar conclusion is drawn from Figure 7. The strong scaling properties for the encoder-decoder architecture used in the MT task are illustrated. The model size ranges from 80 layers, to 320 layers. While the are apparent speedups, additional improvements can be obtained using alternative algorithmic parameters for layer-parallel. We provide practical guidance for parameter selection by analyzing the impact of the number of levels ($L$), coarsening factor ($c_f$), and transformer depth ($N$) on the parallel scaling. To this end, we consider the MC example with layer-parallel configured to perform two forward and one backward iteration. Figure 8 shows that the scalability improves with an increasing number of levels (left image) and larger coarsening factors (middle). However, taking a large coarsening factor can have an impact on the convergence rate (Falgout et al., 2014; Dobrev et al., 2017). The last panel (right) show the benefits of layer-parallel training improve with network depth.

## 5 CONCLUSION

We demonstrated training of neural ODE based transformer models using a layer-parallel approach based on MGRIT. This algorithm uses inexact computations of the gradients for training in order to expose additional parallelism over the layer dimension. This form of layer-parallelism is compatible with other approaches, like data or tensor parallelism. Scalability, parallel speedups using multiple GPUs, and training accuracy are demonstrated for three natural language processing benchmarks when compared to standard serial training. Additionally, the approach naturally distributes large memory loads across multiple GPUs allowing for training of very deep transformer models.

We also explored the impact of inexact gradients resulting in a statistically biased gradient. Layer-parallel training matched serial training in early stages of pre-training. However for some problems, inexact gradients eventually led to diverging or stagnant training dynamics. We corrected this by developing indicators to detect the divergence or stagnation and then transition to a serial gradient computation. We anticipate this to motivate the development of new inexact approaches exposing greater parallelism for training large transformers. Future work will focus on improving MGRIT convergence and the implementation details to include more vectorization while reducing overheads.

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

## A MGRIT - FCF Relaxation

As discussed in Section 3.2.1, the MGRIT algorithm exposes additional parallelism that allows distribution of layers across multiple compute devices (e.g. GPUs). Figure 2 presents the structure of a two level scheme. However for a full picture of MGRIT, we now provide details about the relaxation operator $\mathbf{S}_0$, which is needed for MGRIT efficiency and scalability. To this end, we focus on the finest level of a neural ODE based transformer architecture. Following the notation used in Section 3.2.1, our network has $N_0$ time-steps, coarsened at a rate of $c_f$. The value $c_f$ implicitly defines "coarse point" layers, i.e., the layers $[0, c_f, 2c_f, 3c_f, \ldots]$, which go on to form the first coarse grid. All other points are "fine points."

Forward propagation is denoted as

$$\mathbf{Z}_{n+1} = \Phi(\mathbf{Z}_n) \text{ for } n = 0 \ldots N_0 - 1, \tag{5}$$

where $\Phi$ is a discrete forward propagator implementing encoder-decoder (Eq. 3) or encoder/decoder-only (Eq. 1) architectures. This section explicitly discusses only forward propagation. Backward propagation is similar, but uses an adjoint operator in the propagation step to move backward through the time domain.

The relaxation which forms the action of $\mathbf{S}_0$ consists of two phases. The first one is fine-relaxation (F-relaxation), whereby Equation 5 is used to propagate $\mathbf{Z}_n$ starting from each coarse point until, but not including, the following coarse point. Mathematically this is expressed as

$$\mathbf{Z}_{n+c_f-1} = \underbrace{\Phi \circ \Phi \circ \ldots \circ \Phi}_{c_f-1 \text{ times}} \circ \mathbf{Z}_n \tag{6}$$

where $n$ is a coarse point (an integer multiple of $c_f$). By construction, F-relaxation can be applied with $N_0/c_f$ way parallelism. The second phase is the coarse-relaxation (C-relaxation). This takes the final step

$$\mathbf{Z}_{n+c_f} = \Phi(\mathbf{Z}_{n+c_f-1}) \tag{7}$$

for each of the $N_0/c_f$ coarse points. Here again, the available parallelism is $N_0/c_f$.

Combining these two phases by first applying F-relaxation, then C-relaxation, and finally F-relaxation again, yields FCF-relaxation. Again, this can be computed with $N_0/c_f$ way parallelism. In Falgout et al. (2014), the authors show experimentally that the use of FCF-relaxation is needed for multilevel scalability, yielding a similar result to multigrid reduction in space. In Dobrev et al. (2017), the need for FCF-relaxation is further developed theoretically for linear model problems, where FCF-relaxation is shown to damp error over large parts of the spectrum effectively (i.e., eigenvalues of the spatial discretization with magnitude away from 1, often corresponding to oscillatory error). FCF-relaxation is presented formally in Algorithm 1. At the start of the algorithm $\mathbf{Z}_n$ contains a guess for the features, while at the conclusion, the features are updated with an improved approximation with reduced high frequency errors (in time). The operator $\mathbf{S}_0$ is then defined as mapping from the initial guess for each $\mathbf{Z}_n$, to the updated, improved features. The parallel-for blocks indicate where the relaxation scheme exposes parallelism.

## B Buffer layers

The convergence of MGRIT in the linear case is determined by the stability function of the time-stepping scheme and the eigenvalues of the time-stepping matrix (Dobrev et al., 2017). In the case of MGRIT's application to neural network pretraining, one is therefore interested in the stability constraints arising from the Euler time-stepping scheme due to the feed-forward structure of modern neural networks. To investigate this further, we propose examining the Lipschitz constants arising from pretraining, which can be easily estimated even for transformers.

As motivation, suppose we are solving

$$\frac{dy(t)}{dt} = f(y(t)),$$

**Algorithm 1** FCF-Relaxation

1:  **procedure** FCF-RELAXATION($\mathbf{Z}$)
2:      **parallel-for** $k \leftarrow 0$ to $N_0/c_f - 1$ **do**                    ▷ Begin first F-relaxation
3:          $n \leftarrow c_f k$
4:          **for** $i \leftarrow 0$ to $c_f - 2$ **do**                    ▷ Do $c_f - 1$ propagation steps
5:              $\mathbf{Z}_{n+i+1} \leftarrow \Phi(\mathbf{Z}_{n+i})$
6:          **end for**
7:      **end parallel-for**
8:      **parallel-for** $k \leftarrow 1$ to $N_0/c_f$ **do**                    ▷ Begin first C-relaxation
9:          $n \leftarrow c_f k$
10:         $\mathbf{Z}_n = \Phi(\mathbf{Z}_{n-1})$
11:     **end parallel-for**
12:     **parallel-for** $k \leftarrow 0$ to $N_0/c_f - 1$ **do**                    ▷ Begin second F-relaxation
13:         $n \leftarrow c_f k$
14:         **for** $i \leftarrow 0$ to $c_f - 2$ **do**
15:             $\mathbf{Z}_{n+i+1} \leftarrow \Phi(\mathbf{Z}_{n+i})$
16:         **end for**
17:     **end parallel-for**
18: **end procedure**

for some given initial condition (a simplification of Equation (4)). An Euler iteration would be $y^{n+1} = y^n + \Delta t f(y^n)$, and let the error be $e^n = y(t_n) - y^n$. Recalling the usual Taylor's theorem, along with it's assumptions of smoothness, $y(t_{i+1}) = y(t_i) + \Delta t f(y(t_i)) + O(\Delta t^2)$, one can write the error relation as

$$e^{i+1} = e^i + \Delta t(f(y(t_i)) - f(y^i)) + O(\Delta t^2)$$
$$= e^i + \Delta t(f'(y^*)e^i) + O(\Delta t^2) = (1 + \Delta t f'(y^*))e^i + O(\Delta t^2),$$

where we used the mean value theorem to produce $y^*$. Thus, by the recursive nature, if $(1 + \Delta t f'(y^*)) \gg 1$, then any initial error would amplify and cause massive issues for MGRIT when using Euler's method.

Based on this heuristic, we propose examining the largest eigenvalue of the Jacobian of the individual layers, which can indicate *which* layers are causing divergence when applying MGRIT. Calculating the Jacobian itself for a self-attention or MLP layer is computationally intractable given the large sequence length and hidden dimension. This leads us to empirically estimate the Lipschitz constant via a Monte Carlo approach, which is tightly correlated with the intended value (Paulavičius and Žilinskas, 2006).

In Figure 9, we show the estimated Lipschitz constants during the training of a GPT2 decoder network in the usual serial fashion. Remarkably, as the network trains and becomes more expressive, the rate of change of the Lipschitz constant at each layer is not uniform. It appears that the last few layers change significantly first, followed by the initial layers, while the middle layers remain stagnant for longer. It is known that the gradient updates are greater in magnitude (Wang et al., 2024) for the deeper layers, but we cannot explain the rise in the Lipschitz constant for the early layers. We note that the change in the Lipschitz constant is unrelated to the change in the magnitude of the weights themselves, $\frac{\|w - w_0\|}{\|w_0\|}$ where $w$ is the weights at a specific iteration while $w_0$ the initial weights, as shown in Figure 10.

Regardless of the mechanics driving the change in the Lipschitz constant, the prescription for MGRIT is clear: create "buffer" layers where the first and last layers are computed serially, targeting exact computation of the layers with large estimated Lipschitz constants. MGRIT will perform layer-parallel computations on the middle portion, where the estimated Lipschitz constants are more modest. In other words, a few transformer layers are moved to the open/close layers in Figure 1 from the ParallelNet. Besides moving the layers, we also tweaked the $\Delta t$ (e.g. $h$ from Equation (3)) for the open/close layers: we simply give them $\Delta t = 1$ for the open/close layers, while the transformer layers in the ParallelNet will have the typical $\Delta t = \frac{1}{L}$ where $L$ is the number of layers in the ParallelNet. This method results in greatly increased alignment between the serial and parallel runs in decoder-only networks, as shown in the Figure 11.

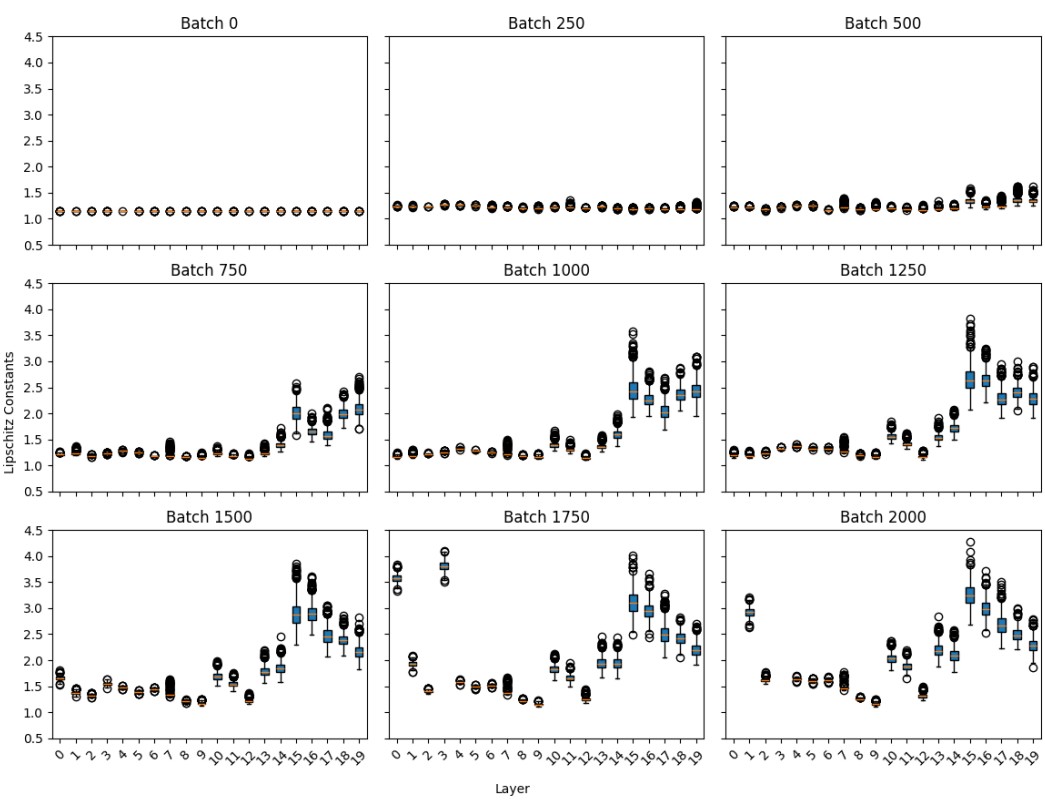

Figure 9: Plots illustrating the Lipschitz constants of each layer as one trains a GPT2-decoder network. Note that the last few layers are the first to change, followed by the initial transformer layer.

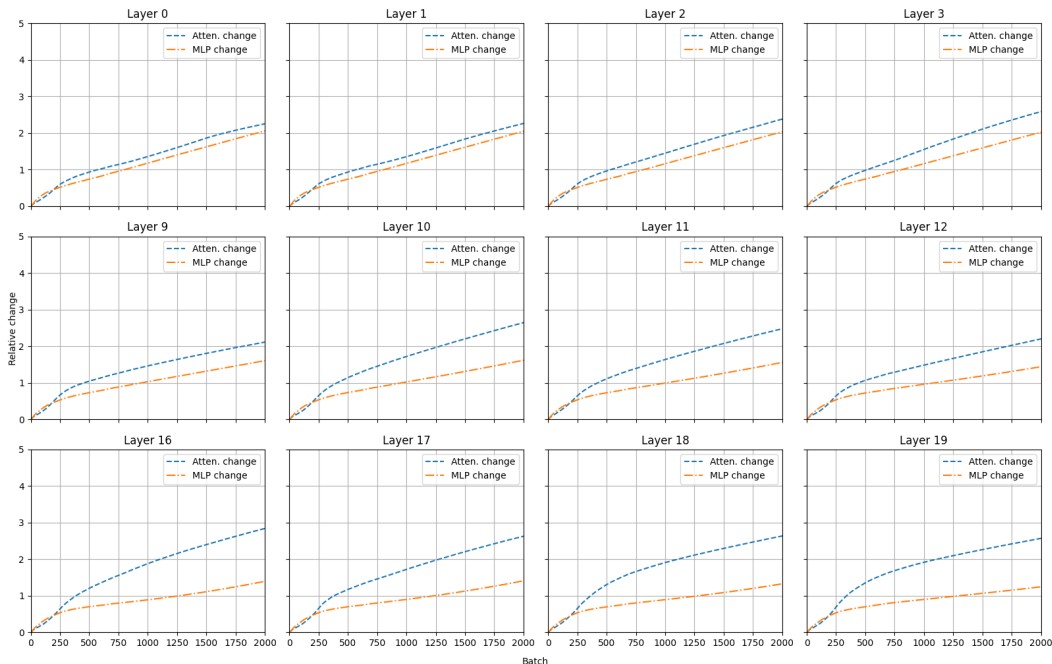

Figure 10: Plots illustrating the changes in relative weight values during the training of a GPT decoder-only network, with transformer weights broken down into attention and MLP components. While all layers clearly change, the impact on the Lipschitz constant is not direct.

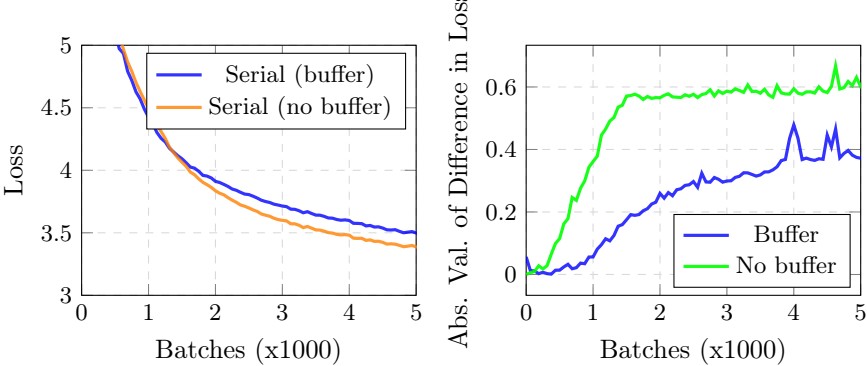

Figure 11: (Left) Loss plots of training GPT-2 decoders in serial for two different configurations. The buffer indicates four of the twenty layers are in the open/close layer (two each) with the remaining sixteen in the middle with $\Delta t = 1/16$. The no buffer indicates all the layers are in the middle with $\Delta t = 1/20$. There is no significant difference in loss between the serial versions of the two configurations. (Right) The absolute difference between the two serial runs against their corresponding layer parallel runs. While the serial dynamics are similar, note that having the buffer layers significantly improves the layer parallel dynamics.

## C    HYPERPARAMETER AND EXPERIMENTAL SETUP

The implementation uses the layer-parallel software TorchBraid (Cyr et al., 2025) built on PyTorch. Experiments and scaling studies are conducted on the HPC systems: Jean-Zay GPU nodes, each equipped with eights V100 and 720 GB of memory and Singra GPU

| Parameter / Example | BERT | MC | ViT | MT | GPT |
|---|---|---|---|---|---|
| Batch-size $B$ | 32 | 8 | 4 | 8 | 256 |
| Dim. feed-forward | 3072 | 128 | 3072 | 2048 | 3072 |
| Dropout | 0.1 | - | - | 0.1 | - |
| Max. length $L$ / Patch size (ViT) | 224 | 2048 | 16 | 274 | 1024 |
| Optimizer | AdamW | SGD | Adam | Adam | AdamW |
| Model dimension $d$ | 768 | 128 | 768 | 512 | 768 |
| Num. heads $H$ | 12 | 1 | 12 | 8 | 12 |
| Num. encoder layers $N_{\mathrm{Enc}}$ | 128 | 4 | 32 | 6 | - |
| Num. decoder layers $N_{\mathrm{Dec}}$ | - | - | - | 6 | 20 |

Table 2: Transformer hyperparameters used for all pretraining problems.

| Parameter / Example | BERT | MC | ViT | MT | GPT |
|---|---|---|---|---|---|
| Step size $h$ | 1 | 1 | 1 | 1 | 1 |
| Num. levels | 2 | 2 | 2 | 2 | 2 |
| Num. forward iterations | 1 | 2 | - | - | - |
| Num. backward iterations | 1 | 1 | 1 | 3 | 1 |
| Coarsening factor | 4 | 8 | 4 | 3 | 4 |
| Pre-smoothing relaxation | F | F | F | F | F |

Table 3: The configuration details of the strong scaling experiments. A dash in the forward iterations indicates serial in the forward solve.

compute nodes, consisting of dual AMD EPYC 7513 Processors and a single A100 80GB GPU.

Table 2 specifies the hyperparameters for the transformers used to generate the results presented in Section 4. For all except the BERT, the parameters of the transformer layers are initialized using PyTorch's default initialization. For the BERT initialization, we follow the pre-LN initialization scaling detailed in (Wang et al., 2024) that provides enough stability for us to train the extremely long BERT network without gradient collapse. In particular, the MLP, value, and output projections of the transformers are scaled by $\sqrt{\log 2L}$. The BERT data is preprocessed using a masked-language modeling with 20%, which is higher than the original BERT manuscript, but has been found to be useful in more recent implementations (Liu et al., 2019).

Table 3 reports the MGRIT configuration used for the strong scaling experiments described in Section 4. Additionally, Table 4 summarizes the hyperparameter values used for tuning the MT task, obtained using Bayesian optimization. Table 5 shows the hyperparameters for the GLUE finetuning; we remark that finetuning on the 128 layers BERT model proved to be challenging, but we only seek to compare the serial versus the parallel-switching training procedures.

Finally, we note that dropout is often used for regularization, however, the layer parallel paradigm cannot simply adopt the `Dropout` classes in existing software. This is because the layers corresponding to exactly $c_f$ (e.g. layers 1, 3, 5, 7 in Figure 2) must have the same masks while doing the relaxation and the coarse solve to ensure the iterations will converge. As such, we implemented a solution whereby the masks do not update unless explicitly specified by the user.

| Hyperparameter | Values |
|---|---|
| Model dimension | $512, 1024$ |
| Dropout | $0.1, 0.3$ |
| Gradient accumulation | $1, 4, 16$ |
| Parameter initialization | Torch's default, Xavier |
| Number of warming steps | $2000, 4000, 8000$ |
| Tokenization | Spacy, BPE, Unigram |
| Vocabulary size | $8000, 32000$ |

Table 4: The range of hyperparameters used during Bayesian optimization for the MT task.

| Parameter / Example | CoLA | MRPC | QNLI |
|---|---|---|---|
| Batch size | 8 | 16 | 16 |
| Weight decay | 0.01 | 0.01 | 0.01 |
| Learning rate | 3e-5 | 2e-5 | 2e-5 |
| Warm up steps | 100 | 0 | 0 |

Table 5: Parameters for finetuning the subset of GLUE benchmarks for the large BERT model. All optimization is performed using PyTorch implementation of AdamW. All computations are done in FP16 precision. Note that we only seek to compare the performance of the pure serial versus the parallel-switching model, and not the exact values.

