# OpenReview forum: "Layer-Parallel Training for Transformers"
_ICLR.cc/2026/Conference — Submitted to ICLR 2026_

### Official Review · Reviewer_8f2R · 2025-10-20

**Soundness:** 2
**Presentation:** 3
**Contribution:** 2
**Rating:** 2
**Confidence:** 3

**Summary:**

This work develops an MGRIT-based strategy for transformer training, enabling parallelism over layers. Layer-parallel training achieves speedups and memory reduction on multiple GPUs. The paper introduces a methodology which allows the training to switch back to serial training when the gradients bias becomes too large. The authors evaluates its impact on pre-training, accuracy, and fine-tuning with language model datasets.

**Strengths:**

1. The authors conduct experiments on various tasks, including BERT pre-training, text classification, vision transformer, machine translation and GPT2 pre-training.

2. The paper is well written.

**Weaknesses:**

1. Although the paper includes extensive experiments across different tasks, there is a lack of a comprehensive evaluation of the model performance. For example, only some loss curves are provided (see Figure 4), but the y-axis spacing is quite large. Small differences in loss could have a significant impact on these tasks.

2. I noticed in the appendix that most tasks use a very small batch size, with only 32 for BERT pre-training. This is far smaller than the batch size typically required for actual pre-training.

**Questions:**

Provide comprehensive evaluation of the model performance, especially acc. on downstream tasks.

---

### Official Review · Reviewer_KVCY · 2025-10-27

**Soundness:** 3
**Presentation:** 3
**Contribution:** 2
**Rating:** 4
**Confidence:** 4

**Summary:**

This paper introduces a layer-parallel training approach for Transformer models based on the neural ODE formulation and the MGRIT method. To correct inexact gradients, the authors propose monitoring the divergence of residuals between iterations and switching to serial training when this indicator stagnates. The method is verified across various models, benchmarks, and hyperparameters, demonstrating its effectiveness.

**Strengths:**

- The paper is well structured and easy to follow.
- The method is evaluated across multiple model architectures and hyperparameter settings.
- Monitoring residuals effectively identifies the transition point between parallel and serial training.

**Weaknesses:**

- The main contribution of the paper is the application of an existing layer-parallel training method to encoder–decoder architectures and the introduction of a monitoring mechanism for training transition. However, applying the layer-parallel approach to encoder–decoder architectures does not appear to differ significantly from its application to encoder-only or decoder-only architectures, making the contribution rather limited.
- In Figure 2, when training at level-1, Layers 1, 3, 5, and 7 are placed on Device 1, but they are distributed across Devices 1 and 2 at level-0. If parameter transmission is required at every training step, this would introduce significant overhead.
- The paper lacks comparisons with existing layer-parallelism methods in multi-GPU training experiments.
- Appendix C omits key pre-training parameters. Given the different iterations and switching strategies between training methods, details such as learning rate and scheduling are necessary.
- Writing issues: Some key methodological details are missing, such as GPU implementation (line 259) and backward propagation (line 265). These are critical for understanding the approach and should be briefly introduced despite page limitations.

**Questions:**

See weaknesses

---

### Official Review · Reviewer_YbRf · 2025-10-28

**Soundness:** 2
**Presentation:** 3
**Contribution:** 2
**Rating:** 2
**Confidence:** 3

**Summary:**

The paper introduces a layer-parallel training method for transformers.
By viewing the transformer's depth as the time axis of a neural ODE, the authors apply the MGRIT algorithm to parallelize computations over the layer dimension, i.e., forward and backward passes in different layers can run in parallel.
Because MGRIT produces inexact gradients, this method is switched off and reduced to none based on an indicator.
Experiments on BERT, GPT-2, ViT, and machine-translation models show the accuracy and speedups achieved by the proposed method.

**Strengths:**

The idea is interesting. Although the methods come from numerical ODEs and are not new, applying them to training transformers is conceptually interesting and orthogonal to other parallel strategies.

**Weaknesses:**

- Unfair comparison baseline: The serial baseline is single-GPU sequential training, while the proposed method uses multi-GPU resources. Instead, standard multi-GPU baselines like TP or PP should be compared.
- Limited layer-parallel training horizon: Once switched to serial mode, the algorithm never re-enables MGRIT. Moreover, the horizon where MGRIT is enabled is very limited. In the GPT-2 pretraining experiment, MGRIT is disabled after 1,000 batches, corresponding to 0.25B tokens (batch size = 256, sequence length = 1024, as shown in Appendix B). This horizon is very short, and training on billions or trillions of tokens is common today. Overall, this method seems unable to scale in current manuscript.

**Questions:**

- How does the method behave when combined with standard TP/PP on equal GPUs?
- Could a bidirectional (serial <-> parallel) switching strategy yield further speedup?

---

### Official Review · Reviewer_RbvQ · 2025-11-01

**Soundness:** 3
**Presentation:** 3
**Contribution:** 3
**Rating:** 6
**Confidence:** 3

**Summary:**

This paper proposes a layer-parallel training paradigm for transformers, building on a neural ODE viewpoint to enable multilevel parallel-in-time algorithms that parallelize computation across the layer dimension. By leveraging the MGRIT algorithm, both forward and backward passes of transformers are made parallel, significantly enhancing scalability as model depth increases. The approach introduces gradient bias due to inexact solutions, for which the authors develop adaptive control mechanisms that switch to serial training or adjust the parallelization parameters when necessary. The methodology is validated on BERT, GPT, ViT, and machine translation models, demonstrating speedups with minimal loss of accuracy compared to conventional serial training, and showing negligible impact on downstream fine-tuning performance.

**Strengths:**

- The paper addresses a core scalability bottleneck in transformer models, layerwise serialization, by proposing a layer-parallel strategy that is compatible with large-scale distributed training.

- The methodology includes an insightful and actionable bias/gradient control mechanism, allowing adaptive switching between inexact (layer-parallel) and exact (serial) computation as training approaches convergence—a pragmatic solution to bias-induced optimization pitfalls.

- Experiments are substantial: Results span diverse architectures (BERT, GPT, ViT, MT) and include strong scaling, convergence fix mechanisms, and ablation studies on parameters like number of MGRIT levels and coarsening factor.

**Weaknesses:**

- **Insufficient Baseline Coverage in Scaling Experiments**: In the scaling studies (Figures 6–8), while classic strong scaling is shown for different architectures, there is no direct comparison to pipeline or tensor parallelism baselines under identical hardware and problem settings. It is not clear how much of the observed speedup is unique to the proposed method, as opposed to achievable by state-of-practice alternatives.
- **Ambiguity in Practical Integration**: The presentation lacks substantial discussion of integration overheads when combining the layer-parallel scheme with other parallelization paradigms (data, tensor, or pipeline parallelism) in realistic, large-cluster deployments. The claim that layer-parallel is "fully compatible" is not experimentally demonstrated.
- **Limited Exploration of Hyperparameters**: While some space is allocated to tuning, the interplay between MGRIT hyperparameters and problem/task type (BERT, GPT, ViT, MT) is not extensively or systematically studied, and guidance remains empirical rather than principled.

**Questions:**

1. Can the authors provide a more systematic quantitative comparison (wall-clock time, speedup, GPU utilization, memory footprint) between layer-parallel training and standard pipeline/tensor/data parallelism baselines, especially under identical hardware setups?
2. What are the main challenges in integrating layer-parallel training with other forms of parallelism (data/pipeline/tensor) at scale? Are there observable overheads or compatibility pitfalls?

---

### Meta-Review · Area_Chair_Xw8o · 2026-01-06

**Summary:**

The AC has carefully reviewed the paper independently to ensure a fair decision, even in the absence of a rebuttal. Overall, reviewers question the fairness of the baselines, the scalability and practical integration of the proposed method, the limited novelty relative to prior layer-parallel approaches, and the insufficient methodological and performance evaluations needed to assess real-world applicability. A summary of the reviewers' main concerns and questions follows:

- Reviewer RbvQ: insufficient baseline coverage in scaling experiments (no direct comparison to pipeline or tensor parallelism baselines under identical hardware and problem settings), ambiguity in practical integration in realistic, large-cluster deployments, and limited exploration of hyperparameters.

- Reviewer YbRf: unfair comparison baselines with respect to single- and multi-GPU experiments, and a limited layer-parallel training horizon in terms of sequence length.

- Reviewer KVCY: limited contribution beyond encoder–decoder architecture applicability, lack of comparisons with existing layer-parallelism methods in multi-GPU training experiments, and some writing issues.

- Reviewer 8f2R: lack of a comprehensive evaluation of model performance and a limited experimental setup, including a very small batch size (only 32) for BERT pre-training.

**Reviewer Concerns:**

No rebuttal was provided; therefore, nothing could be addressed.

**Reviewer Scores:**

This paper received one acceptance-inclined score and three rejection-inclined scores. Although some reviews were of low quality, the authors did not respond to certain reviewer comments, leaving no opportunity to address those concerns and increase the scores. Therefore, the score remains unchanged.

---

### Decision · Program_Chairs · 2026-01-26

Reject